# Use of epigenetically modified bacteriophage and dual beta-lactams to treat a *Mycobacterium abscessus* sternal wound infection

Madison Cristinziano[1], Elena Shashkina[2,3], Liang Chen [2,3], Jaime Xiao[4], Melissa B. Miller[5], Christina Doligalski[4,6], Raymond Coakley[7], Leonard Jason Lobo[7], Brent Footer[8], Luther Bartelt [8], Lawrence Abad[1], Daniel A. Russell[1], Rebecca Garlena[1], Michael J. Lauer [1], Maggie Viland[1], Ari Kaganovsky[1], Emily Mowry[1], Deborah Jacobs-Sera [1], David van Duin[8], Barry N. Kreiswirth [2,3] ✉, Graham F. Hatfull [1] ✉ & Anne Friedland[8] ✉

Nontuberculous mycobacterium (NTM) infections are challenging to manage and are frequently non-responsive to aggressive but poorly-tolerated antibiotic therapies. Immunosuppressed lung transplant patients are susceptible to NTM infections and poor patient outcomes are common. Bacteriophages present an alternative treatment option and are associated with favorable clinical outcomes. Similarly, dual beta-lactam combinations show promise in vitro, but clinical use is sparse. We report here a patient with an uncontrolled *Mycobacterium abscessus* infection following a bilateral lung transplant and failed antibiotic therapy. Both smooth and rough colony morphotype strains were initially present, but treatment with two phages that kill the rough strain – including epigenetic-modification to overcome restriction – resulted in isolation of only the smooth strain. The rough and smooth strains have similar antibiotic susceptibilities suggesting that the phages specifically eliminated the rough strain. Dual beta-lactam therapy with meropenem and ceftazidime-avibactam provided further clinical improvement, and the phages act synergistically with meropenem in vitro.

Nontuberculous mycobacterium (NTM) infections are a significant cause of morbidity and mortality among people with cystic fibrosis and other underlying structural lung abnormalities, and immunosuppressed patients, including solid organ transplant (SOT) recipients[1–3]. The incidence of NTM infections appears to be rising[4], and clinical outcomes are often poor[5,6]. *Mycobacterium abscessus*, a rapid-growing NTM species, is the second most common cause of NTM infection. The *M. abscessus* complex consists of three genomically characterized subspecies: *M. abscessus* subsp. *abscessus*, *M. abscessus* subsp. *bolletii* and *M. abscessus* subsp. *massiliense*. Subspecies *abscessus* and *bolletii*

[1]Department of Biological Sciences, University of Pittsburgh, Pittsburgh, PA, USA. [2]Center for Discovery and Innovation, Nutley, NJ, USA. [3]Hackensack Meridian School of Medicine, Nutley, NJ, USA. [4]University of North Carolina School of Medicine, Chapel Hill, NC, USA. [5]Department of Pathology and Laboratory Medicine, University of North Carolina, Chapel Hill, NC, USA. [6]University of North Carolina School of Pharmacy, Chapel Hill, NC, USA. [7]Division of Pulmonary Diseases and Critical Care Medicine, University of North Carolina, Chapel Hill, NC, USA. [8]Division of Infectious Diseases, University of North Carolina, Chapel Hill, NC, USA. ✉e-mail: Barry.Kreiswirth@hmh-cdi.org; gfh@pitt.edu; aefriedl@email.unc.edu

contain an inducible macrolide resistance gene, *erm (41)*, whereas subspecies *massiliense* lacks this gene and is susceptible to macrolides[7]. Although *M. massiliense* and *M. bolletii* could be united to form one subspecies, genome comparisons are consistent with the *M. abscessus* complex comprising three separate entities[8–10]. *M. abscessus* is very challenging to treat due to its intrinsic multi-drug resistance and ability to form biofilms. Treatment is prolonged with complex, multi-drug regimens, and often adjunctive surgery for management of extrapulmonary disease[3]. Patients often fail treatment, relapse or develop drug-associated toxicities[5]. In contrast to immunocompetent patients, almost half of NTM infections in SOT recipients are extrapulmonary[2,11] and can present with subtle findings, including non-healing wounds, small papules, or cutaneous sinus tracts with non-purulent drainage[12].

There is clearly a need for new therapeutic approaches for NTM infections[13]. Several case studies have shown favorable outcomes with bacteriophage therapies[14–18], although few therapeutically useful phages are available, and the host ranges are unpredictable. Thus, each clinical isolate must be tested for its phage susceptibility profiles prior to consideration of a personalized clinical intervention[14]. The narrow phage repertoire and strain variability in phage infection profiles thus limit broader usage[19,20].

The beta-lactams imipenem and cefoxitin are recommended as part of the American Thoracic Society (ATS) guidelines for the treatment of *M. abscessus* infections[21]. Beta-lactams target enzymes that regulate peptidoglycan synthesis, including L-D transpeptidases, D-D transpeptidases, and carboxypeptidases, which are crucial for cell wall synthesis. However, *M. abscessus* has a chromosomally encoded beta-lactamase (Bla$_{Mab}$) which reduces the efficacy of beta-lactams. The potential synergy of beta-lactams used in combination was first observed by Pandey et al., who found that ceftazidime (which is otherwise inactive against *M. abscessus*) lowers the MICs of imipenem and ceftaroline by 16 to 64-fold[22]. Larger studies of various combinations of dual beta-lactams have shown similar synergy[23]. Dual beta-lactam therapy has been previously reported in three patients, all of whom achieved culture clearance[24–26].

The mechanism of dual beta-lactam synergy is poorly defined, but may be due to either a) an increase in the total number and selection of enzymes inhibited by using two drugs in comparison to a single drug or b) drug-binding conformationally changing in the target sites of the enzymes, allowing for binding of the second drug, which disrupts in vitro enzyme activity[22,23]. Not all beta-lactamase inhibitors bind to Bla$_{Mab}$. However, the diazabicyclooctanes inhibitors including avibactam bind to Bla$_{Mab}$ and increase susceptibility to β-lactam antibiotics. Avibactam has been shown to increase the activity of numerous antibiotics, including amoxicillin, ceftaroline, cefuroxime, ertapenem, tebipenem, and imipenem in vitro[23].

Here we report a complex case of a lung transplant recipient with *M. abscessus* sternal osteomyelitis and soft tissue infection, with relapsed/refractory infection despite aggressive medical and surgical management. The patient was treated with two phages, including one with epigenetic modifications, and dual beta-lactam therapy. To our knowledge, this is the first case of treatment of extrapulmonary NTM disease with either dual beta-lactams or the use of epigenetically modified phages.

## Results

### A patient with a refractory NTM infection

A male in his 60s with a history of idiopathic pulmonary fibrosis underwent a bilateral orthotopic lung transplant. Five months post-transplant, he developed pain and erythema over his sternal wound. His symptoms failed to improve with courses of levofloxacin/doxycycline and amoxicillin/clavulanate. He subsequently developed superficial papules over the surgical wound site, and 1 month later, a swab of one of these papules grew macrolide-resistant *M. abscessus* on bacterial culture (Table 1 and Fig. 1A, Day 0). Microbiological assays indicated that the cultures contained mixtures of both rough and smooth colonies, and after purification, these were designated as *M. abscessus* GD276A and GD276B, respectively (Table 1).

Two days post-diagnosis (pd), he underwent wide debridement down to the sternum with the removal of the screws and sternal plate. Sternal plate culture, bone culture, and chest soft tissue cultures grew *M. abscessus*. Two days later, he underwent additional debridement of the anterior sternum and soft tissue, but soft tissue and sternal margins remained positive. He was initially treated with imipenem, azithromycin, eravacycline, and linezolid, and briefly with amikacin which

## Table 1 | *M. abscessus* susceptibilities (MIC, μg/ml) to antimicrobials

| Strain | GD276 | GD276-2 | GD276-1 | GD276p1 | GD276p2 | GD276p3 | GD276p4 |
|---|---|---|---|---|---|---|---|
| **Morphotype** | Mixed[a] | Smooth | Rough | Smooth | Rough | Smooth | Smooth |
| **Source** | Chest swab | Sternum | Sternum | Chest swab | Chest swab | Chest swab | Skin Biopsy |
| Amikacin | 16 (S)[b] | 16 (S) | 16 (S) | 16 (S) | 16 (S) | 16 (S) | 32 (I) |
| Bedaquiline | 0.06 | ND | ND | 0.12 | 0.12 | 0.12 | 0.12 |
| Cefoxitin | 64 (I) | 32 (I) | 128 (R) | 128 (R) | 32 (I) | 32 (I) | 32 (I) |
| Ciprofloxacin | >16 (R) | >4 (R) | 4 (R) | >4 (R) | >4 (R) | >4 (R) | >4 (R) |
| Clarithromycin | 16 (R) | 16 (R) | >16 (R) | >16 (R) | >16 (R) | >16 (R) | >16 (R) |
| Clofazimine | 0.25 | 0.25 | 0.25 | 0.25 | 0.12 | 0.25 | 0.5 |
| Doxycycline | >8 (R) | >8 (R) | >8 (R) | >8 (R) | >8 (R) | >8 (R) | >8 (R) |
| Eravacycline | ND | ND | ND | 0.12 | ND | 0.12 | 0.12 |
| Imipenem | 16 (I) | 8 (I) | 16 (I) | 16 (I) | 8 (I) | 16 (I) | 16 (I) |
| Linezolid | 16 (I) | 16 (I) | 16 (I) | 4 (S) | 16 (I) | 16 (I) | 16 (I) |
| Moxifloxacin | >4 (R) | >4 (R) | >4 (R) | >4 (R) | >4 (R) | >4 (R) | >4 (R) |
| Omadacycline | ND | ND | ND | 0.5 | 0.5 | 0.12 | 0.12 |
| Tedizolid | ND | ND | ND | ND | ND | 4 | <0.6 |
| Tigecycline | 0.25 | 0.12 | 0.25 | 0.25 | 0.25 | 0.25 | 0.25 |
| TMP-SMX | >4 (R) | 4 (R) | >4 (R) | 4 (R) | >4 (R) | >4 (R) | >4 (R) |

[a]Sample contains both rough and smooth colony morphotype, which for phage testing were separated and designated GD276A and GD276B, with rough and smooth morphotypes, respectively.
[b]MICs (μg/ml) are shown for each strain and antibiotic, with R, S, and I indicated resistant, sensitive, and intermediate phenotypes.

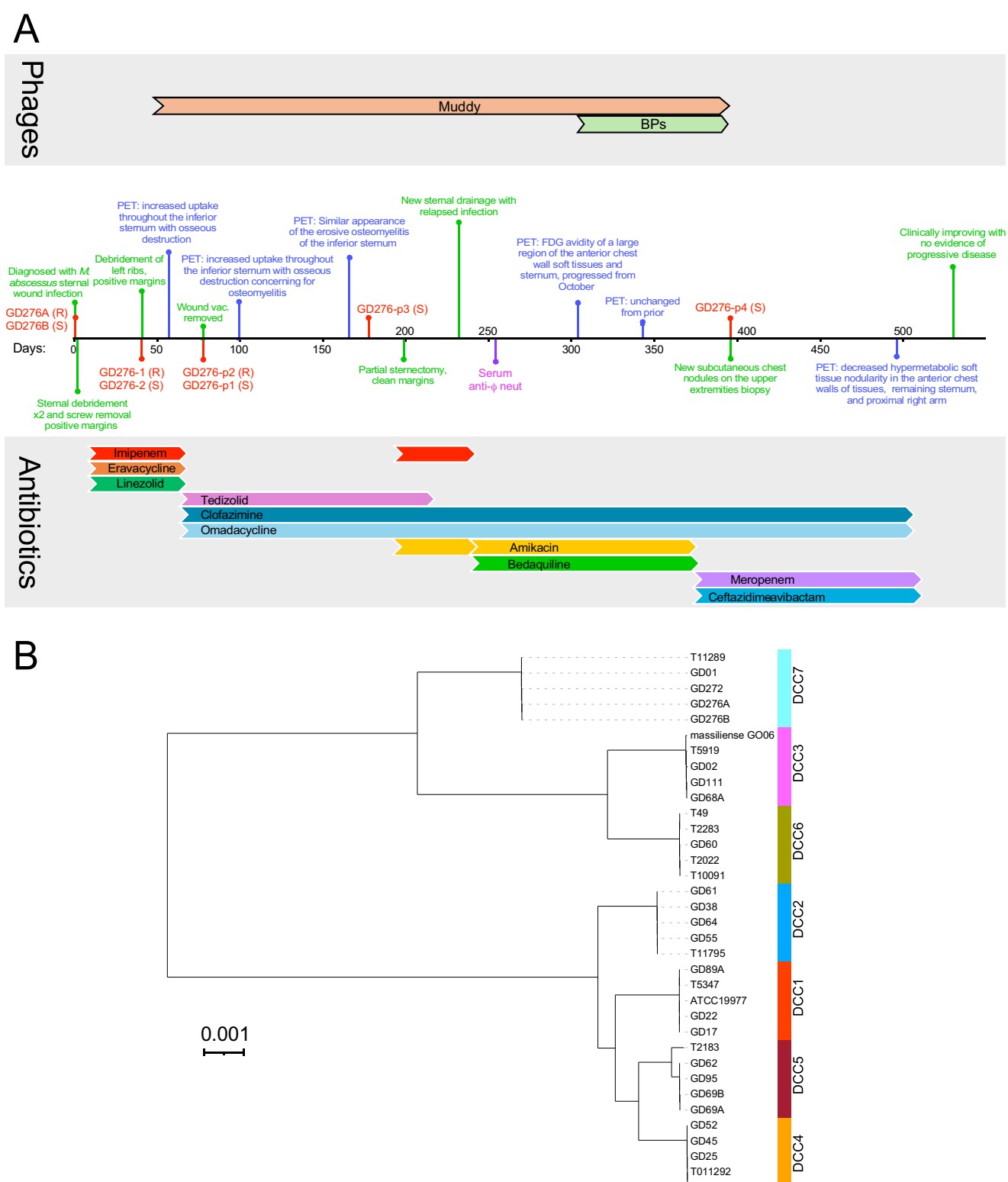

**Fig. 1 | Timeline of case study and strain comparisons. A** A timeline is shown of the course of treatments for the patient described in this study, with day 0 corresponding to the time of diagnosis and intervals of each 50 days post-diagnosis (pd) shown. Key events are noted, including the times of strain isolations (shown in red text), PET imaging (shown in blue text), and other clinical events (shown in green text). The duration of antibiotic and phage treatments (colored for distinction) are shown below and above the timeline, respectively. **B** Phylogenetic relationships of *M. abscessus* strains indicating dominant circulating clones (DDC) 1 to 7 as described previously[50]. Strains GD276A, GD276B, and GD272 are all members of DDC7.

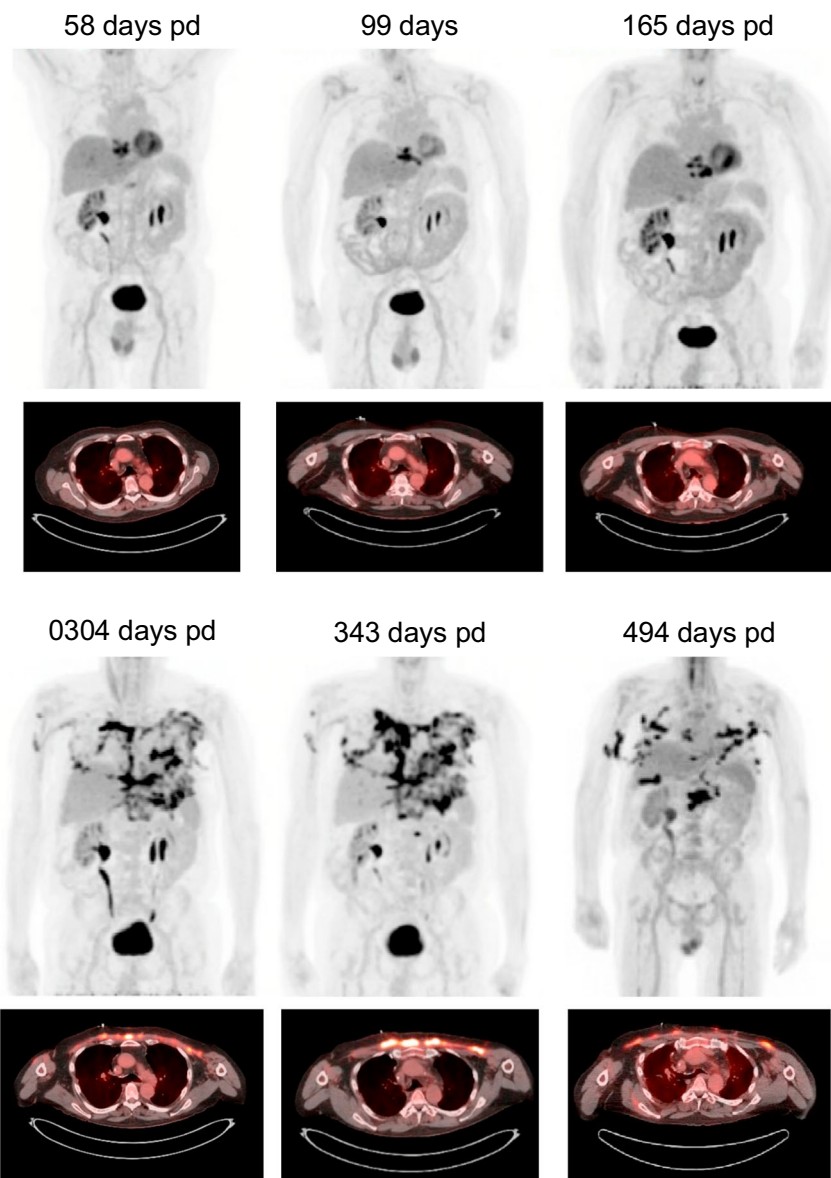

**Fig. 2 | PET-CT scans.** The patient's PET-CT scans are shown over the course of his infection, ordered by the number of days post-diagnosis. The top images show the maximum intensity projection (MIP) images from a coronal view. The bottom images show the F-18 Fluorodeoxyglucose (FDG) uptake in the chest wall from an axial view.

was stopped within 48 hours due to an increase in creatinine levels (Fig. 1A). He was discharged from the hospital after susceptibility results returned on clofazimine, tedizolid and omadacycline with a wound vacuum in place (Table 1 and Fig. 1A).

He was re-admitted 40 days pd with bubbling around the wound vacuum and concern for pleural fistula. He underwent repeated debridement of the sternal wound and two exposed ribs (Fig. 1A). Rib cultures were positive for *M. abscessus*. PET-CT scans are shown in Fig. 2.

### *M. abscessus* GD276A and GD276B are related to dominant circulating clone 7 strains

Strains GD276A and GD276B were sequenced to completion using a combination of sequencing reads using Illumina and Nanopore technologies. GD276A has a 5,026,869 bp genome and two plasmids, designated pGD276A-1 and pGD276A-2 (Table S1), which are 92,821 bp and 41,369 bp, respectively. GD276B contains identical plasmids to these, and an identically sized genome, although it differs from GD276A with ~12 single nucleotides polymorphisms (SNPs) although about half of these are in a putative homing endonuclease gene

(GD276A_0306). Of the remaining SNPs, two are in intergenic regions, and two more are non-synonymous SNPs in individual genes. However, one of the SNPs is a single base insertion in gene GD276A_04285 such that GD276B codes for a wild-type copy of the *mps1* gene, and GD276A has a frameshift mutation. The *mps1* gene is involved in glycopeptidolipid (GPL) synthesis, and cell wall GPLs are characteristic of smooth colony *M. abscessus* isolates; frameshift mutations in *mps1* and *mps2* are commonly associated with a rough colony morphotype, and the 1 bp insertion in the GD276A gene is thus likely responsible for the colony morphotype. We also fully sequenced a separate clinical isolate, strain GD272, which has a 5,026,669 bp genome, and is very similar to GD276A/B, differing from GD276B by 23 SNPs, and two regions of small (1–2 bp) insertions/deletions; it is a rough strain and has a SNP in the *mps1* translation initiation codon (changing GTG to GCG).

A conserved sequence phylogenetic comparison shows that GD276A and GD276B are members of dominant circulating clone (DCC) 7, together with other strains in subspecies *masilliense,* including GD01 (Fig. 1B and Fig. S1) which was the target of a previously reported phage treatment[16]. Two other strains—GD58 and GD272—are also in DCC7, both of which were isolated from patients in the same

hospital as the patient reported here. Both plasmids found in GD276A are present in GD58, while GD272 contains an identical copy of pGD276A-1 and an ~18 kbp plasmid identical to pGD276A-2 albeit with a 21 kbp deletion. Strains GD58, GD272, GD276A, and GD276B all contain two prophages, grouped in MabA1 and MabG[20,27], respectively (which are now included in Actinobacteriophage clusters HA and HG, respectively[28]). The MabG prophages are designated prophiGD58-1, prophiGD272-1, prophiGD276A-1, and propGD276B-1, and the MabA1 prophages are designated prophiGD58-2, prophiGD272-2, pro-phiGD276A-2, and prophiGD276B-2. The MabG prophages are notable in that they carry phage-encoded ESX-secreted toxin (PEST) systems in which the polymorphic toxins include a domain related to the tuberculosis necrotizing toxin (TNT)[27–29].

## Determination of phage susceptibility profiles

Strains GD276A and GD276B were screened using a targeted phage panel similar to that described previously[14–16,19]. Although thousands of *M. smegmatis* phages have been isolated and sequenced[30], only a few infect any strain of *M. abscessus*[19,20], and GD276A was found to be fully susceptible to phage Muddy and a host range derivative (Muddy_HRM[GD04]), as well as the closely related phage, Maco6 (Fig. 3A). Strain GD276A is not susceptible to either BPs (or its derivatives) or ZoeJ, both of which efficiently infect the GD01 strain and were used therapeutically[16]. The smooth strain GD276B did not show efficient infection by any of the phages, although very turbid infectious regions are seen with phage Muddy and its relatives (Fig. 3A), consistent with phage susceptibility profiles reported previously for smooth strains[18].

Although phage BPs (and its derivatives) do not efficiently plaque on GD272 or GD276A, we observed some plaques at high phage titers, which could be either host range mutants, or epigenetically modified derivatives escaping a host restriction system. To discern between these, plaques were picked from the GD272 plate and re-plated onto *M. smegmatis* and GD272, where they showed similar efficiencies of plaquing (EOPs) (Fig. 3B). Plaques were then picked from the *M. smegmatis* plate and similarly retested where they plaqued with strongly reduced EOP on GD272 relative to *M. smegmatis* (Fig. 3B). These observations suggest that the inefficient plaquing of BPs on GD272 results from a restriction-like system, and this was confirmed by propagating a high titer BPs lysate on GD272 and demonstrating its efficient plaquing on both GD272 and *M. smegmatis* (Fig. 3C). Because GD272 and GD276A are closely related we also tested to see if, when propagated on GD272, BPs also efficiently plaques on GD276A, which it clearly does (Fig. 3C). However, this BPs lysate forms only very turbid areas of infection on GD276B, similar to phage Muddy (Fig. 3A).

To determine the therapeutic suitability of phage Muddy for treating the GD276 infection, we combined serial dilutions of GD276A and GD276B with serial dilutions of Muddy and incubated in a liquid culture prior to plating on a solid medium to monitor for bacterial killing (Fig. 3D). Muddy efficiently killed GD276A over a wide range of concentrations and multiplicities of infection (MOI) (Fig. 3D), but poor killing of GD276B was consistent with the plaque phenotype (Fig. 3A, D). We also tested for bacterial survivors following a challenge of $2 \times 10^8$ CFU's bacteria with Muddy at an MOI of 10 (Fig. 3E). Killing of GD276B was inefficient as anticipated, and GD276A was killed well, although survivors were observed at a prevalence of about $10^{-4}$ (Fig. 3E).

## Expanding phage therapeutic options

Phage Muddy shows an infection and killing profile that should make it suitable for therapeutic use on a compassionate-use basis. The proportion of GD276A survivors following a Muddy challenge is higher than desirable, and the GD276B smooth strain is hardly killed at all. However, other cases of mixed rough/smooth infections have shown some clinical improvement with a reduction of the rough strain following phage treatment[15], making therapy with Muddy a reasonable

option. In principle, a lytic derivative of BPs could also be used therapeutically, but propagating large quantities of phage on the pathogenic GD272 (or GD276A) strain is considerably more challenging than using *M. smegmatis* as a host. We, therefore, investigated the basis for the restricted plaquing of BPs on GD272 and GD276A with the goal of constructing a recombinant strain of *M. smegmatis* for propagating epigenetically modified phage for therapeutic application. First, we examined the GD272 and GD276A genomes for restriction-modification systems, and identified an *hsdSMR*-like locus present in both strains (Fig. 4A) but absent from other strains such as GD01[16] and GD82[31] that are fully sensitive to BPs (Fig. 4B). This *hsdSMR* system is part of a ~20 kbp genomic island that is seemingly integrated by integrase-mediated site-specific recombination using an *attB* site in a host tRNA-Phe gene (MAB_t5015 in *M. abscessus* ATCC19977) (Fig. 4A). This genomic island contains other genes implicated in phage defense, including a CBASS system and a TIR-domain protein (Fig. 4A).

To test if the *hsdSMR* locus restricts BPs growth, we constructed recombinant strains expressing either the entire *hsdSMR* locus or just the specificity and modification (*hsdS*, *hsdM*) genes (pMC02 and pMC09, respectively) (Fig. 4A). Both plasmids transform *M. smegmatis* efficiently, but neither substantially change the EOP of BPs, suggesting that this locus is not involved in defense against BPs, or perhaps one or more of the genes are not properly expressed in this strain. However, we observed that if we propagate BPs on the *M. smegmatis*pMC09 strain, the phage behaves similarly to when it is propagated on GD272 and infects GD272 and GD276A much more efficiently (Fig. 4C). The *M. smegmatis*pMC09-derived phage preparation (BPsD*33*HTH_HRM10[pMC09]) kills strain GD276A and GD276B with a similar profile to that seen with Muddy (Figs. 3D, E, 4D, E); challenge with the epigenetically modified BPs results in reasonably good killing of the GD276A strain but with a substantial number of survivors (Fig. 4E) and poor killing of the smooth strain, GD276B.

## Phage resistance in vitro

To determine the mechanisms giving rise to resistance to phage Muddy, four colonies surviving a phage challenge were purified and characterized (Fig. 5). Muddy was shown to have reduced EOP on all four derivatives, although some infection was observed at high viral titers (Fig. 5A). Three of the Muddy-resistant mutants had reverted to a smooth colony morphotype, reflecting a resistance mechanism reported previously[20]. Complete DNA sequencing of the Muddy-resistant rough strain (RMM3; Fig. 5A) showed that it has a single base insertion frameshift mutation in the *rpoZ* gene (MAB_2822c in the *M. abscessus* ATCC19977 reference strain). Although the role of *rpoZ* in phage infection is unclear, we previously reported a 4 bp deletion in *rpoZ* causing a frameshift mutation in a Muddy-resistant derivative of *M. abscessus* GD19[20]. Once the modified preparation of BPs was available, we also isolated and purified four survivors of BPsΔ*33*HTH_HRM10[pMC09] infection (Fig. 5B). Two of these had similarly reverted to a smooth colony morphotype, and sequencing of one (RMB09-1) confirmed that the single base insertion mutation (at coordinate 4,230,042 in GD276A) causing a frameshift in *mps1* conferring the rough colony morphotype in GD276A had reverted to the wild-type *mps1* sequence present in GD276B (Fig. 5C). We also sequenced one of the rough BPs-resistant mutants which has a single base insertion in an intergenic region between divergently transcribed genes (MAB_2775c and MAB_2776 in the ATCC19977 reference strain), and presumably influences the expression of one or both genes. Phage-resistant mutations have not previously been reported at this locus, and the functions of both genes are poorly defined.

## Patient response to phage therapy

The patient was admitted ~58 days pd; (Fig. 1A) for initiation of treatment with phage Muddy, the only candidate available at that time. Following approval of an eIND application by the FDA and local IRB

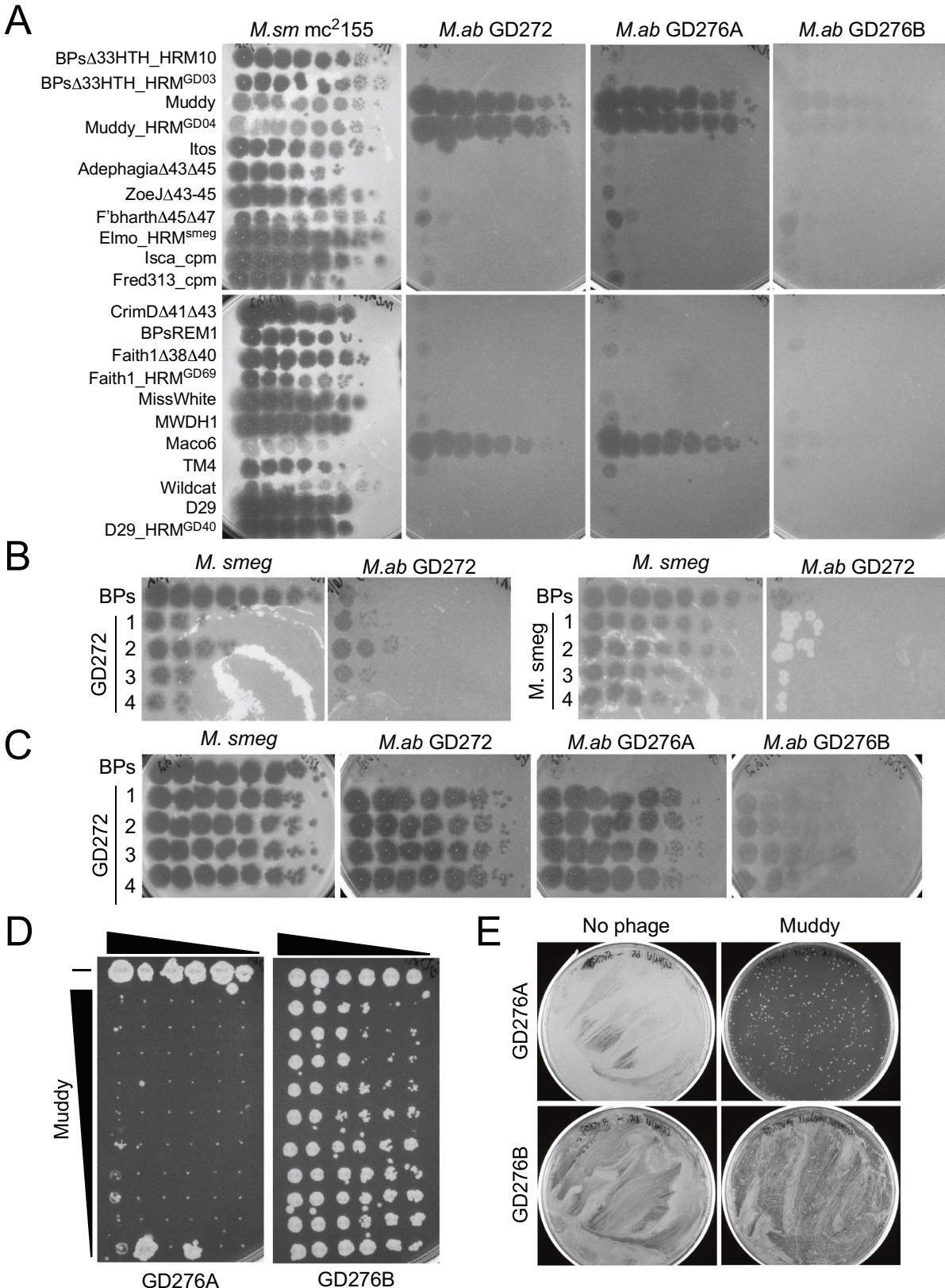

permission, the phage was administered intravenously twice daily at a dose of $10^9$ PFU, mirroring the administration route, dosage, and frequency as used in prior compassionate-use interventions for NTM infections[15,16,31]. A baseline PET scan obtained during that admission, prior to phage initiation, showed a soft tissue defect overlying the inferior sternum with increased radiotracer uptake within the soft

tissues and underlying sternum with osseous destruction of the sternum and costochondral junctions concerning osteomyelitis.

The phage was tolerated well, with no serious adverse events (SAEs). An *M. abscessus* culture collected immediately prior to phage therapy again showed the presence of both rough and smooth colony morphotypes (designated GD276-1 and GD276-2, respectively). After

**Fig. 3 | Phage infection profiles of *M. abscessus* GD276A and GD276B. A** Twenty-two phage candidates were tested for infection of three clinically isolated strains of *M. abscessus*. Phage lysates were tenfold serially diluted and spotted onto lawns of *M. smegmatis* mc²155, and *M. abscessus* strains GD272, GD276A, and GD276B. **B** Left panels: Four individual plaques (#1-4) of BPsΔ33HTH_HRM10 were picked from a lawn of GD272 (similar to that shown in panel A) and re-plated on *M. smegmatis* and GD272. The top row labeled 'BPs' is the lysate of BPsΔ*33*HTH_HRM10 grown on *M. smegmatis*. Right panels: A plaque from each of the four lysates spotted on *M. smegmatis* in the left panel were picked, propagated on *M. smegmatis,* and tenfold serial dilutions were plated on *M. smegmatis* and GD272. The experiment shows that the ability of BPs to infect GD272 is not heritable, and thus is a phenotypic phenomenon in which epigenetic modification is exclusively derived from growth on GD272, but not from *M. smegmatis*, and therefore overcomes restriction by GD272 when propagated on this strain. **C** High titer lysates of BPsΔ33HTH_HRM10 from the four purified plaques shown in (**B**) were propagated on GD272, tenfold serially diluted, and plated on *M. smegmatis*, GD272, GD276A, and GD276B. **D** Strains GD276A and GD276B were tenfold serially diluted and incubated with either no phage (top row shown with "−") or with tenfold serially diluted Muddy (rows 2–9) with row two containing $10^9$ PFU. **E** Cultures of GD276A ($2.4 \times 10^8$ cells) and GD276B ($2.7 \times 10^8$ cells) were mixed with $2.4 \times 10^9$ and $2.7 \times 10^9$ of Muddy, respectively, and incubated in liquid culture for 2 days. Then, 0.1 mL was plated for the growth of survivors on a solid medium. Plates were incubated for 11 days at 37 °C.

purification and retesting, these were shown to have identical sensitivity profiles as the original GD276A and GD276B isolates. Strains GD276-1 and GD276-2 (and presumably GD276A and GD276B) have similar antibiotic profiles, with the exception of a modest difference in cefoxitin susceptibility.

His wound vacuum was removed 77 days pd, but he had ongoing sternal drainage with cultures positive for smooth and rough *M. abscessus* colony morphotypes (designated GD276p1 and GD276p2, respectively). These strains exhibit the same phage infection profiles and similar antibiotic susceptibilities of the earlier isolates (Fig. 6A). A follow-up PET scan obtained after 6 weeks of phage therapy (99 days pd) showed re-demonstration of increased uptake throughout the inferior aspect of the sternum with associated osseous destruction concerning for osteomyelitis, but with some improvement in uptake in the adjacent soft tissues.

He had ongoing sternal drainage from various sternal wound sinus tracts and continued to feel poorly. A subsequent PET-CT scan 165 days pd showed a similar appearance of soft tissue infection and erosive osteomyelitis involving the sternum and surrounding sterno-chondral joints (Fig. 2). An *M. abscessus* culture (GD276p3) collected in clinic from a swab of his chest drainage (175 days pd) grew only the smooth colony morphotype with a phage susceptibility profile similar to *M. abscessus* GD276B (Fig. 6A).

He was re-admitted 200 days pd for partial sternectomy. Numerous margins were collected, and all were clear histologically with negative cultures. At this point, he developed severe neuropathy in his bilateral feet, thought to be due to tedizolid. He was discharged on amikacin, imipenem, clofazimine, and omadacycline and continued Muddy phage therapy.

The patient initially did well post-operatively, but a month later (~230 days pd) presented with a new sternal sinus tract that tested culture positive for *M. abscessus* (Table 1 and Fig. 1A). He developed some dizziness and unsteadiness, which resolved with imipenem cessation and bedaquiline was added to the regimen of amikacin, clofazimine, omadacycline, and phage Muddy (Fig. 1A). Serum collected at 255 days pd showed little or no neutralization of phage Muddy, although this was not surprising due to administration of basiliximab (Fig. 6B). Follow-up imaging with non-contrast CT ~250 days pd showed widespread soft tissue disease without drainable fluid collections. Thoracic surgery was consulted, but it was believed the soft tissue disease was too widespread for surgical resection. A follow-up PET scan obtained 304 days pd showed significant progression of disease (compared to PET on day 165) with the increased area, size, and avidity of uptake within the anterior chest wall and soft tissues centered at the sternectomy site and extending along the bilateral pectoralis flaps without drainable fluid collection.

At this time, the epigenetically modified BPs phage (BPsΔ*33*HTH_HRM10$^{pMC09}$) was added to this therapy regimen at the same dosage and regimen as for Muddy (Fig. 1A). The phage was again very well tolerated without SAEs. The patient was discharged on both phages, amikacin, bedaquiline, omadacycline, and clofazimine. A PET scan obtained six weeks later (343 days pd) showed no significant changes from prior, with significant uptake in the anterior chest wall tissues, sternum, and proximal right upper arm (Fig. 2).

The patient continued to feel poorly and had ongoing drainage from his sternal wound and prior chest tube sites. At 390 days pd, he developed new subcutaneous chest nodules, and a biopsy culture was positive for *M. abscessus* (Fig. 1A). The recovered *M. abscessus* isolate (GD276p4) has a smooth colony morphology, and is poorly infected by any of the phages, similar to the GD276B and GD265-p3 isolates (Fig. 6A). With apparent depletion of the rough colony morphotype strain the phage therapy was discontinued at that time (~400 days pd).

### Successful dual beta-lactam therapy

Although dual beta-lactam therapy has only been reported in a few clinical cases, in vitro studies show a very promising profile[22,24,32–34]. Given the lack of improvement in his current therapy, after susceptibility testing (Table 2) and based on prior usage, dual beta-lactam therapy with meropenem and ceftazidime-avibactam was initiated (~415 days pd); imipenem was avoided as the patient poorly tolerated it. Amikacin and bedaquiline were stopped, and he continued on omadacycline and clofazimine. The patient experienced significant improvement in his energy levels and a dramatic decrease in the sternal wound and prior chest tube site drainage soon after starting dual beta-lactams. A PET-CT obtained ~490 days pd and about 3 months after initiation of dual beta-lactams showed an overall decrease in hypermetabolic soft tissue nodularity over the anterior chest wall tissues, remaining sternum and proximal right arm, suggestive of improving myositis/osteomyelitis (Fig. 2). He continued dual beta-lactam therapy and tolerated it well without substantial adverse reactions through to ~560 days pd, with no evidence of progressive disease.

**Synergy between phages and meropenem.** Retrospectively, we examined whether the combination of phages with the dual beta-lactam treatment could have synergistic benefits, as has been reported for other pathogens[35–37]. Testing synergy was not readily feasible with the *M. abscessus* GD276A rough strain as phages Muddy and BPsΔ*33*HTH_HRM10$^{pMC09}$ both kill the strain efficiently over a broad range of concentrations and MOIs (Figs. 3, 4). However, understanding synergy with the GD276B smooth strain is more relevant to this clinical case, as only the smooth strain was present when the beta-lactams were administered. Although phage administration was stopped at that time, enough phage particles could have remained at the site of the infection to exert an influence. Although the phages alone show the very poor killing of the GD276B smooth strain, synergistic activity with the beta-lactams is of interest and could impact the clinical course.

Using fixed concentrations of avibactam (4 μg/ml), we varied the meropenem and concentrations of either phages Muddy or BPsΔ*33*HTH_HRM10$^{pMC09}$ (Fig. 6C and Fig. S2). We observed substantial synergistic effects and more efficient killing of *M. abscessus* GD276B in the presence of either phages with the effect amplified at higher phage concentrations (Fig. 6C), and in conditions where the phages show minimal inhibition of bacterial growth when grown alone. We estimate

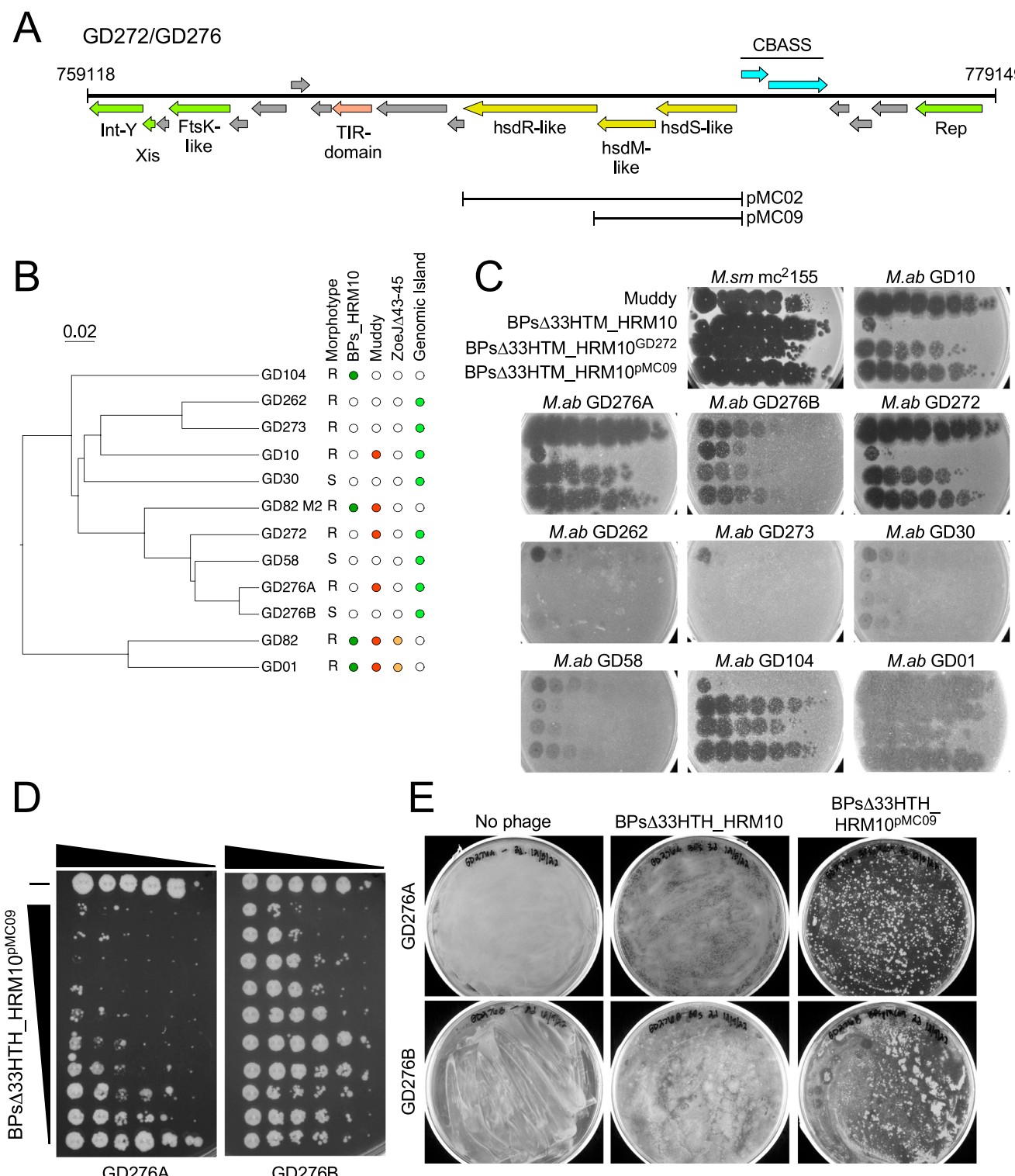

that the reduction in meropenem is approximately fourfold in the presence of $10^7$ PFU phage relative to the absence of phage.

## Discussion

We describe here a challenging clinical case of an *M. abscessus* infection and experimental therapeutic strategies. Unfortunately, cases of this type are not uncommon, and a high proportion of NTM infections are intrinsically antibiotic resistant, have acquired resistances, and are non-responsive to aggressive antibiotic therapies[38]. Solid organ transplant recipients are at notable risk due to their need for lifelong

immunosuppression[2]. This patient was treated for over 18 months and developed numerous drug-associated toxicities and relapsed on treatment despite partial resection of the sternum with clean margins.

In vitro studies of phage-*Mycobacterium* interactions in model systems show promise[39–41], and several cases of compassionate-use phage therapies for *Mycobacterium* infections have been reported, with many favorable outcomes, although not consistently so[15–18,31]. There are notable limitations to the broader use of phages to control NTM infections, and the paucity of therapeutically useful phages and the highly variable spectrum of phage sensitivity profiles are

**Fig. 4 | A genomic island containing an *hsdSMR*-like restriction system.**
**A** Organization of a 20,031 bp genomic island present in some DCC7 strains, including GD272, GD276A, and GD276B. Green-colored genes are phage-related, pink and aqua-colored gene are defense-related, and the yellow genes are the *hsdSMR* locus that was cloned into recombinant plasmids: pMC02 and pMC09.
**B** Phylogenetic relationships of several DCC7 strains and their colony morphotype, susceptibility profiles to phages BPsΔ33HTM_HRM10, Muddy, and ZoeJΔ43-45, and the presence of the genomic island shown in (**A**). Rough and smooth morphologies are indicated by "R" and "S," respectively. Efficient phage infection of each strain is shown by a filled colored circle, and lack of infection by an empty circle. Strains containing the genomic island are indicated by a bright green circle. **C** Plaque assays of DCC7 strains with phages Muddy, BPsΔ33HTM_HRM10, BPsΔ33HTM_HRM10 propagated on *M. abscessus* GD272

(BPsΔ33HTM_HRM10^GD272), and BPsΔ33HTM_HRM10 propagated on *M. smegmatis* mc²155 carrying the recombinant plasmid pMC09 (BPsΔ33HTM_HRM10^pMC09). Phage lysates were tenfold serially diluted and spotted onto lawns of *M. smegmatis* mc²155 and the DCC7 strains. **D** Killing assays of strains GD276A and GD276B with phage BPsΔ33HTM_HRM10^pMC09. GD276A and GD276B were tenfold serially diluted and incubated with either no phage (top row shown with "−") or with tenfold serially diluted BPsΔ33HTM_HRM10^pMC09 (rows 2−9) with row two containing $10^9$ PFU of BPsΔ33HTM_HRM10^pMC09. **E** Cultures of GD276A ($2.0 \times 10^8$ cells) and GD276B ($3.3 \times 10^8$ cells) were mixed with $2.0 \times 10^9$ and $3.3 \times 10^9$ PFU of BPsΔ33HTM_HRM10^pMC09, respectively, and incubated in liquid culture for 3 and 2 days, respectively. Then, 0.1 mL was plated for the growth of survivors on a solid medium. Plates were incubated for 11 days at 37 °C.

particularly problematic[14,20,24]. In addition, most in vitro studies are performed to kill bacteria in planktonic growth rather than with biofilms. Little is known about the role of NTM biofilms in vivo, although both smooth and rough colony morphotypes of *M. abscessus* form biofilms in vitro[42]. Understanding the interactions between phages and NTM biofilms both in vitro and in vivo would be helpful. Investigations of phage activity in animal model systems would also be informative[40].

Screening of the clinical isolates revealed only one phage (Muddy) with potential efficacy, and only for the rough colony morphotype strain. Although phage resistance has not been observed in prior cases where only a single phage was used for NTM infections, resistance can be detected in vitro, and readily so for the strains in this case. We determined here that the basis for poor infection by a second phage (BPs and its lytic derivatives) was a host restriction system, which could be overcome by either propagating the epigenetically modified phage on the host strain, or on a recombinant strain of *M. smegmatis* expressing the modification system. Bioinformatic analyses suggest the modification is likely an N-6 adenine-specific DNA methylation, and that the system is present in all or most other DCC7 strains but is also present in other strains and other *Mycobacterium* species. The epigenetically modified BPs phage should thus be incorporated in future screening of NTM strains for phage susceptibilities.

The response to the phage treatment is complex because of the involvement of both rough and smooth strains in the infection. Genomic analysis shows that the rough strain (GD276A) is a derivative of the smooth strain (GD276B) with mutational inactivation of GPL biosynthesis, and both Muddy and the modified BPs do not infect the smooth strain. Thus, the expectation was that the treatment could potentially eliminate the rough strain, and confer some clinical benefit, as has been reported in some other cases[15]. It is encouraging that only the smooth colony morphotype of *M. abscessus* was isolated in samples collected approximately four months and nine months after initiation of therapy with Muddy (Fig. 1A). Importantly, because the rough and smooth strains have similar antibiotic susceptibilities, a strong argument can be advanced that phage administration was specifically responsible for clinical depletion of the rough strain. No adverse reactions to the phage therapy were observed, and because the patient already had a PICC line for IV antibiotics, administration was relatively straightforward. We note that the patient was immunosuppressed and thus did not have neutralizing antibodies to the phage after several months of treatment.

*M. abscessus* GD276A appears to readily mutate to give phage-resistant derivatives in vitro, although a prominent mechanism is a simple reversion to the smooth colony morphotype with genetic reversion to the wild-type sequence in strain GD276B. This resistance mechanism has been reported previously[20] and could also have occurred in the patient, although such isolates are indistinguishable from the smooth strain present prior to phage therapy. We also observed previously reported phage resistance in vitro from mutations in *rpoZ*[20]. An additional mutation giving resistance maps to an intergenic region, presumably altering the expression of one or both of the

adjacent genes; the mechanism is unknown and warrants further investigation. No rough strains were isolated up to nine months following the start of phage treatment, suggesting that if such mutants arose in vivo, they had poor fitness.

This case is only the fourth report of clinical treatment of *M. abscessus* with dual beta-lactams, and the first for both treatment of an extrapulmonary *M. abscessus* infection and in an immunosuppressed patient. It is only the second case in a patient with macrolide-resistant disease, which is generally more difficult to treat. Interestingly, the phages appeared effective in reducing or eliminating the rough strain, but not the smooth strain, even though the antibiotic susceptibilities are similar, supporting the interpretation that this results directly from phage-mediated killing. The rough and smooth isolates have slightly different susceptibilities to some of the beta-lactam combinations (Table 2), but there is a striking synergism between both phages and the beta-lactams used therapeutically for the smooth strain (Fig. 6C). This was unknown at the time of treatment in this case, but suggests that the beta-lactams and phages could be combined for effective therapy in other patients.

It is also noteworthy that in the three prior cases, dual beta-lactam therapy was initiated early in treatment along with other antibiotics confounding the interpretation of efficacy. Here, it was initiated only after failure of over a year of treatment with multiple surgeries, a multidrug regimen (including IV amikacin), and phage therapy but resulted in clear clinical and radiographic improvement. Dual beta-lactam combination therapy remains an exciting potential option for difficult-to-treat *M. abscessus* infections, especially in combination with phages when available, although long-term stability, optimal dosing regimens, and potential discrepancies between in vitro susceptibilities and clinical efficacy are potential challenges. Further clinical data are urgently needed[23].

This case illustrates the multiple challenges in treating *M. abscessus* infections. The patient had numerous antibiotic intolerances and developed relapsed disease despite treatment with months of antibiotics and phages and partial sternal excision with clean margins. The incidence of these infections is increasing, and new and better-tolerated therapies are urgently needed. Phage and dual beta-lactam therapies are both promising treatment options which warrant further investigation.

## Methods

### Inclusion and ethics
This research protocol was approved by the University of North Carolina institutional review board (IRB numbers 22-1710, 19-2446) and the US Food and Drug Administration via a single patient expanded access application. The patient provided informed consent, according to CARE guidelines and in compliance with the Declaration of Helsinki principles.

### Bacterial strains
*M. smegmatis* mc²155 is a laboratory strain and was grown as previously described[43]. *M. abscessus* clinical isolates GD40, GD272, GD276A, and

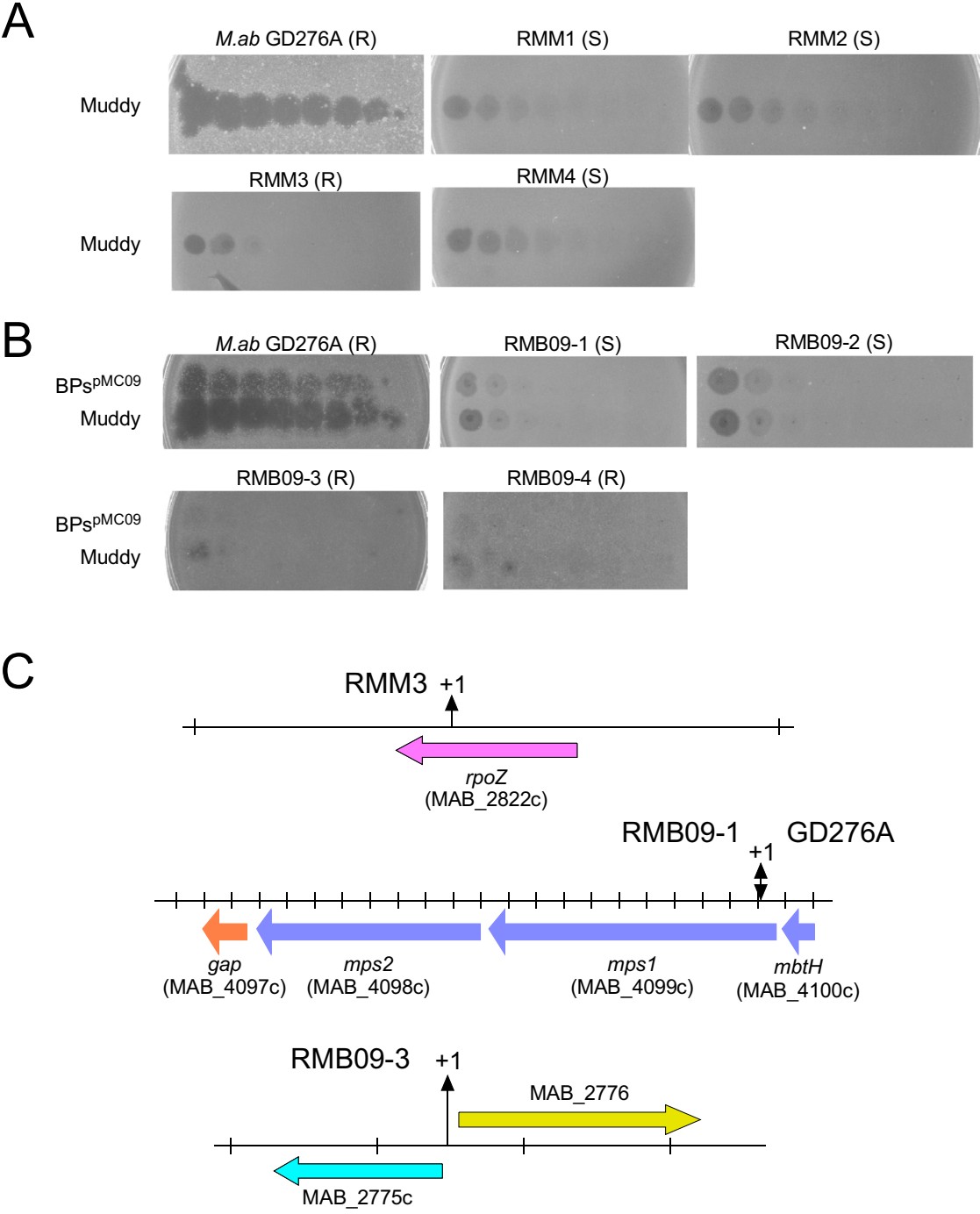

**Fig. 5 | Phage-resistant mutants of GD276A. A** Four Muddy-resistant derivatives of GD276A (RMM1–RMM4) were screened for phage susceptibility to Muddy. The colony morphotype of each strain is shown as rough (R) or smooth (S). **B** Four mutant derivatives of GD276A resistant to BPsΔ*33*HTH_HRM10^pMC09 (RMP09-1 – RMB09-4) were screened for phage susceptibility to Muddy and BPsΔ*33*HTH_HRM10^pMC09. The colony morphotype of each strain is shown as rough (R) or smooth (S). **C** Observed mutations in Muddy or BPsΔ*33*HTH_HRM10^pMC09 resistant strains RMM3, RMB09-1, and RMB09-3. Genes of interest are represented by colored arrows, and single base insertions are indicated by "+1." The top shows the *rpoZ* gene and the location of the +1 frameshift mutation in mutant RMM3. Below that is shown the *mps1-mps2* locus involved in GPL biosynthesis. The double-headed arrow indicates the position where a +1 frameshift mutation is responsible for the rough colony morphotype in GD276A, and where reversion to the wild-type sequence gives the smooth colony morphotype, and phage resistance in mutant RMB09-1. At the bottom is the location of a +1 frameshift mutation in mutant RMB09-3.

GD276B were grown in Middlebrook 7H9 media supplemented with 10% OADC and 1 mM CaCl₂ for 4–5 days, shaking, at 37 °C. For plaque assays and liquid killing assays, *M. abscessus* cultures were sonicated briefly in a cup-horn sonicator (Q-sonica 500) at 30% amplitude with 15 sec on and 10 s off until visibly dispersed.

**Phage and antibiotic susceptibility determination**

*M. abscessus* strains were screened for phage susceptibility using a standard plaque assay to determine the efficiency of plaquing (EOP). A panel of phage candidates were tenfold serially diluted and 3 µl were spotted onto top agar lawns containing 500 µl of *M. smegmatis* mc²155

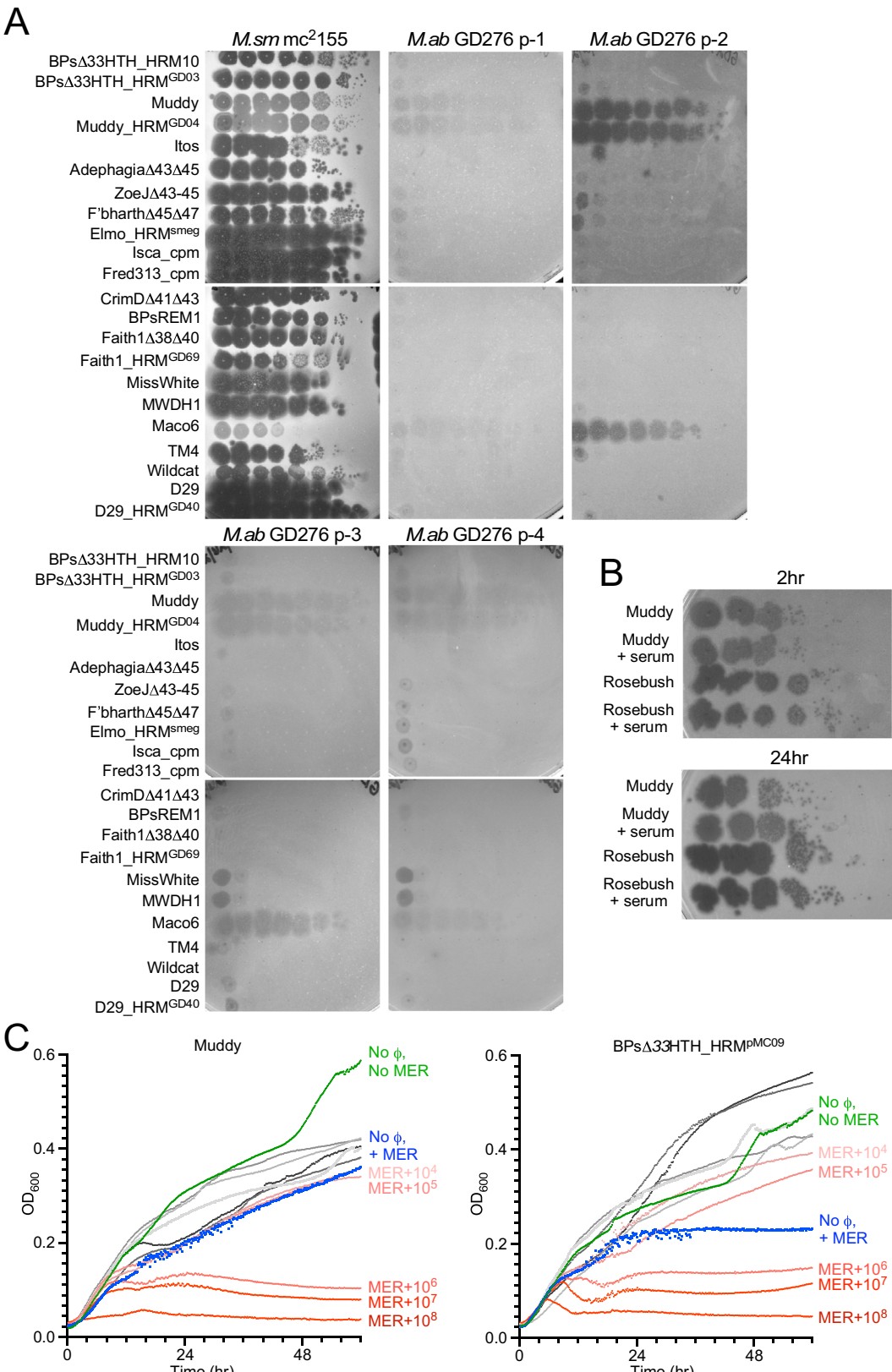

and *M. abscessus* strains. The phages tested were isolated on *M. smegmatis* mc$^2$155 or *M. abscessus* as spontaneously induced prophages[20]. Antibiotic susceptibilities were determined as previously reported[44]. Testing for synergy between phages and the meropenem was performed in 96-well microtiter plates and a BioTek Synergy H1 plate reader and BioTek's Gen5 Data Analysis Software (version 3.10),

using GraphPad Prism (version 10.0) for analysis. Cells ($2 \times 10^7$ CFU) were inoculated into a total of 200 μl volume containing Middlebrook 7H9 medium with 10% OADC and 1 mM CaCl$_2$, with avibactam (4 μg/ml) and varying meropenem concentrations. Either phage Muddy or phage BPs$\Delta$33HTH_HRM10$^{pMC09}$ was added, and the OD$_{600}$ was recorded for 60 h with incubation at 37 °C.

**Fig. 6 | Phage susceptibility profiles of *M. abscessus* strains. A** Plaque assays with tenfold serial dilutions of phages (shown at the left) of *M. abscessus* isolates are shown. The times of isolation of each of the strains is shown in Fig. 1A and Table 1. **B** Phage neutralization assays with patient serum. To test for antibody-mediated phage neutralization, Muddy and an unrelated phage, Rosebush, were mixed with serum collected after 6 months of therapy (see Fig. 1A) and incubated for 2 h (top) and 24 h (bottom), tenfold serially diluted, and spotted onto lawns of *M. smegmatis* mc²155 as indicated. Plates were incubated at 37 °C for 48 h. **C** Antibiotic-phage synergy. *M. abscessus* GD276B (smooth) cultures were incubated with phages Muddy (left) or BPsΔ33HTH_HRM10^pMC09 (right) and antibiotics at 37 °C for 60 h, recording OD_{600}. Cultures grown without antibiotics or phages are shown in green and those with meropenem (2 μg/ml) without phage in blue. Solid gray lines are cultures containing phages and no antibiotics, with the lightest gray containing $10^4$ PFU and the darkest containing $10^8$ PFU. Samples containing both meropenem and phage are shown in red, with the lightest color containing $10^4$ PFU and the darkest color containing $10^8$ PFU, as indicated. Cultures containing meropenem also contained avibactam (4 μg/ml). Data for all readings are shown in Fig. S2 and in the Source Data File.

**Table 2 | Susceptibilities of *M. abscessus* strains to combinations of beta-lactams**

| Strain | R/S[b] | CPT[a] – – | IMI – – | IMI CFT (1) – | IMI AVI (4) – | IMI AVI (4) CAZ (100) | CFT AVI (4) – | CFT CAZ (100) – | CFT AVI (4) CAZ (100) | MER VAB (8) – | CFT MER (1) VAB (8) | MER AVI (4) CAZ (100) |
|---|---|---|---|---|---|---|---|---|---|---|---|---|
| GD276 | R/S | 8[c] | 1 | 0.5 | 1 | 0.5 | 0.5 | 0.5 | 0.25 | 4 | 8 | 1 |
| GD276-2 | S | 32 | 8 | 4 | 2 | 2 | 4 | 0.5 | 0.25 | 4 | 8 | 1 |
| GD276-1 | R | 32 | 8 | 4 | 4 | 0.5 | 1 | 0.5 | 0.25 | 4 | 8 | 1 |

Antibiotic sensitivities are shown as minimum inhibitory (MIC100) concentrations (μg/ml).

[a]Antibiotic abbreviations: *CPT* ceftaroline, *IMI* imipenem, *MER* meropenem, *AVI* avibactam, *VAB* vaborbactam, *CAZ* ceftazidime. Fixed concentrations of drugs are shown in parentheses in μg/ml.

[b]R, S Rough and smooth colony morphotypes; R/S, mixed colony morphotypes

[c]Minimum inhibitory concentration (MIC) values in μg/mls

### Construction of pMC02 and pMC09 recombinant plasmids

Plasmid pMC02 was constructed by amplification of the *hsdSMR* locus of *M. abscessus* strain GD272 using Q5 HiFi 2× master mix (New England Biolabs). The amplicon was purified and cloned into the amplified anhydrotetracycline (ATc)-inducible vector pCCK39[45] using NEBuilder HiFI DNA Assembly master mix (New England Biolabs) and transformed into *E.coli* strain DH5α. pMC09 was constructed in a similar fashion by amplifying the *hsdS* and *hsdM* genes of *M. abscessus* strain GD272, cloning into the amplified pLO74 vector containing the constitutive P_{hsp60} promoter[46], and transforming into *E.coli* strain DH5α.

### Phage neutralization assay

Patient serum samples were used to test antibody-mediated neutralization of the therapeutic phage. Ten μl of phages Muddy and Rosebush containing ~$1×10^7$ PFU/ml and 10 μl of undiluted patient serum were mixed in 80 μl phage buffer (10 mM Tris HCl (pH 7.5), 10 mM MgSO_4, 68.5 mM NaCl, and 1 mM CaCl_2) and incubated at room temperature for two and 24 h. At both timepoints, the phage and serum mixtures were tenfold serially diluted and 3 μl were spotted onto lawns of *M. smegmatis* mc²155. Plates were incubated at 37 °C for 48 h.

### Whole-genome sequencing of bacterial strains

Libraries were prepared with the NEBNext Ultra II FS Library Prep kit (New England Biolabs) and sequenced on an Illumina MiSeq. In the case of GD276A, a library was also prepared for Oxford Nanopore sequencing using a Rapid Barcoding Kit, then run on a MinION. For GD276A, Nanopore reads were assembled using Trycycler[47], the assembly was polished using the Illumina reads, and then it was checked for completeness and accuracy using Consed[48]. For all other strains, Illumina reads were aligned to the parent strain genome using Consed[48], and a custom program (AceUtil) was used to identify differences between the mutant reads and the parent genome. All mutations were confirmed by close inspection of the reads. Whole genome alignments were used to construct phylogenetic trees[19], and prophages were detected using DEPhT[49].

### Therapeutic bacteriophage preparation

Phages for therapeutic use were prepared by amplification and purification by equilibrium density centrifugation[16–18,31]. Phages Muddy and BPsΔ33HTH_HRM10^pMC09 were propagated on *M. smegmatis* mc²155 and *M. smegmatis* mc²155pMC09, respectively, on solid media with the top layer containing 0.35% agar. The top agar was collected and pelleted by centrifugation to remove debris and cells. The supernatant was sterilized by passing through a 0.22 μm filter. The phage was then concentrated by centrifugation at 100,000×*g* for 1 h. The phage pellet was resuspended in phage buffer (68 mM NaCl, 10 mM Tris HCl pH 7.5, 10 mM MgSO_4, 10 mM CaCl_2), and cesium chloride was added to the concentrated phage lysate to create a 4.1 M solution with a density of 1.5 g/mL. A density gradient was created by centrifugation at 132,600×*g* for 16 h. The visible phage band was recovered using a sterile 18 G 1 1/2 in. syringe pierced through the side of the tube and withdrawn in a volume of ~2–3 mL. Phage buffer with 1 mM CaCl_2 and 4.1 M CsCl (density of 1.5 g/mL) was added to the recovered phage particles and subjected to a second round of equilibrium density gradient centrifugation. The visible phage band was collected and stored at 4° C until dialysis. Cesium-banded phage particles were loaded into dialysis cassettes (Slide-A-Lyzer Dialysis Cassettes, 10 K MWCO) and dialyzed against 1 Liter of Ringers Solution (Oxoid Ringers Solution) four times. The dialyzed phage samples were experimentally determined to be less than 190 ppb CsCl by inductively coupled plasma mass spectrometry. Dialyzed phage was sterilized by passing through a 0.22 μm filter. Dialyzed phage was added to sterile Ringers solution and 1.5 M trehalose for a final concentration of 0.5 M trehalose. About 100 μL of the phage-trehalose mixture was added to each sterile glass bottle, a sterile flange cap was added to each vial, and the vials were lyophilized over 36 h and sealed under 0.3 mbar pressure. Endotoxin was shown to be below the level of detection using the EndoZyme II (Hyglos GmbH) assay and USP71 sterility testing was completed by Accugen. Over the course of treatment, vials containing 0.1 mL of Muddy and BPsΔ33HTH_HRM10^pMC09 ranging from $1×10^{10}$ to $1×10^{11}$ PFU were resuspended in 10 mL of Ringers, distributed into syringes, and 1 mL was administered twice daily.

### Statistics and reproducibility

The phage susceptibility screen of *M. abscessus* isolates was performed once, and then therapeutically candidate phages retested; representative data are shown (Figs. 3–6). The *M. abscessus* killing/survival assays (Figs. 3, 4) and neutralization assays were performed once.

## Reporting summary

Further information on research design is available in the Nature Portfolio Reporting Summary linked to this article.

## Data availability

All datasets and materials generated during and/or analyzed during the current study are available from the corresponding author on request without restriction except for clinical details due to privacy laws. Gen-Bank accession numbers of bacterial genomes are as follows: *M. abscessus* GD272 [https://www.ncbi.nlm.nih.gov/nuccore/CP167809] and its plasmids pGD272-1 [https://www.ncbi.nlm.nih.gov/nuccore/CP167810] and pGD272-2 [https://www.ncbi.nlm.nih.gov/nuccore/CP167811]; GD276A [https://www.ncbi.nlm.nih.gov/nuccore/CP167806] and its plasmids pGD276A-1 [https://www.ncbi.nlm.nih.gov/nuccore/CP167807] and pGD276A-2 [https://www.ncbi.nlm.nih.gov/nuccore/CP167808]; GD276B [https://www.ncbi.nlm.nih.gov/nuccore/CP167803] and its plasmids pGD276B-1 [https://www.ncbi.nlm.nih.gov/nuccore/CP167804] and pGD276B-2 [https://www.ncbi.nlm.nih.gov/nuccore/CP167805]; GD276A_RMM3 [https://www.ncbi.nlm.nih.gov/nuccore/CP167794] and its plasmids pGD276A_RMM3-1 [https://www.ncbi.nlm.nih.gov/nuccore/CP167795] and pGD276A_RMM3-2 [https://www.ncbi.nlm.nih.gov/nuccore/CP167796]; GD276A_RMB09_3 [https://www.ncbi.nlm.nih.gov/nuccore/CP167797] and its plasmids pGD276A_RMB09_3-1 [https://www.ncbi.nlm.nih.gov/nuccore/CP167798] and pGD276A_RMB09_3-2 [https://www.ncbi.nlm.nih.gov/nuccore/CP167799]; GD276A_RMB09_1 [https://www.ncbi.nlm.nih.gov/nuccore/CP167800] and its plasmids pGD276A_RMB09_1-1 [https://www.ncbi.nlm.nih.gov/nuccore/CP167801] and pGD276A_RMB09_1-2 [https://www.ncbi.nlm.nih.gov/nuccore/CP167802]. Source data are provided with this paper.

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

## Acknowledgements

We thank Colin Lewis for their excellent technical assistance. This work was supported by grants GM131729 from the National Institutes of Health (G.F.H.), GT12053 from the Howard Hughes Medical Institute (G.F.H.), HATFUL19GO from the Cystic Fibrosis Foundation (G.F.H.), and grant AWD00006613 from Emily's Entourage (G.F.H.). We thank Carlos Guerrero for the discussions.

## Author contributions

J.X., M.B.M., C.D., R.C., L.J.L., B.F., L.B., D.v.D., and A.F. designed the clinical protocol, clinical care and advice, and antibiotic susceptibility testing. M.C., L.A., A.K., M.J.L., E.M., and M.V. screened strains for their phage infection profiles and killing profiles, phage genetics, synergy, and construction of recombinant strains. L.A., D.A.R., R.G., and M.V. sequenced and assembled genomes, comparative genomics, and phylogenetic analyses. E.S., L.C., and B.N.K. did genome analyses and antibiotic susceptibilities for beta-lactam antibiotics. D.J.S., E.M., and M.V. prepared the phages for therapeutic use. D.v.D. and G.F.H. did research design and oversight. The first draft of the paper was written by A.F., M.C., and G.F.H., and was reviewed and edited by all authors.

## Competing interests

The authors declare no competing interests.
