## [Peer Review File · Nature Communications]

REVIEWER COMMENTS

Reviewer #1 (Remarks to the Author):

The work of Cristinziano et al. is sound. It describes the clearance of a rough *M. abscessus* strain from an immunosuppressed patient after the administration of a dual beta-lactam combination with two bacteriophages. The description is clear and detailed. The value of this work lies in the need of knowledge about clinical cases of patients infected with sequenced *M. abscessus* and treated with well-studied phages and an exhaustive in vitro analysis. However, there are some aspects that need to be improved or clarified:

1. In the Introduction section, it should be included the three subsp. of *M. abscessus* explaining that taxonomy is not in consensus right now (with references).
2. It would be necessary to study the in vitro interaction between the antibiotics and the phages to know if a previous in vitro study could have been beneficial for the patient as a guide. It means to perform checkerboard assays or growth OD curves, for instance.
3. It is said in the manuscript that there is no correlation between the effect of phages in vitro and in the patient, but I suggest to include in the Discussion section the following idea: in vitro studies usually are performed to kill bacteria in a planktonic state and no associated to a biofilm. Biofilm assays would be desirable to mimic better the real ambient of the bacteria and phage in the patient.
4. In the Discussion section it would be desirable to add some information about in vivo studies of phage therapy against *M. abscessus* and to comment if they are worthy or not.

As a minor comment, I suggest to check the use of the past and present tense through the manuscript using the past for the results of the authors and the present for known information.

Reviewer #2 (Remarks to the Author):

This is a report of successful treatment of an extrapulmonary NTM infection of a sternal wound using a combined phage therapy/antibiotic approach. While the patient had both rough and smooth *M. abscessus* infections and phages worked only against the former, the addition of dual beta-lactam therapy with meropenem and ceftazidime-avibactam resolved infection. The authors conclude that this therapeutic approach can be successful in complicated NTM skin and soft tissue infections.

This is a well-written and clearly presented study. The conclusions are generally well-supported by the data. While the targeting of NTMs with phages is not especially novel (this group has published much in this space) the context (a sternal wound) and the concomitant use of dual beta-lactam therapy are important and expand the scope of this therapeutic approach.

However, these data raise some questions:

1. Are there any evidence of synergy between the dual beta-lactam therapy and the phages used here against smooth NTM? The authors do not claim that there is but sensitizing effects of phages on otherwise antibiotic resistant bacterial isolates have been noted before and would be important to interrogate here.
2. Is there any evidence that elimination of one of the NTM isolates (the rough strain) made it easier to eliminate the other (the smooth strain) with conventional antibiotics? Polyclonal or polymicrobial infections can be more challenging to clear. It would be interesting to co-culture the two NTM strains to see if this increases the concentration of antibiotic needed to eliminate these relative to treatment of each individually.
3. It would be important to discuss the necessity of the deep susceptibility testing that was done here. Given that the same phages (Muddy, Rosebud) have been used in several of the NTM treatment cases reported by this group are these studies in fact necessary once the presence of a rough strain is established? This is worth asking since the intensive testing regiments probably limit the more widespread and generic adaptation of this approach.

REVIEWER COMMENTS

Responses to reviewer's comments are in **red type** below.

Reviewer #1 (Remarks to the Author):

The work of Cristinziano et al. is sound. It describes the clearance of a rough *M. abscessus* strain from an immunosuppressed patient after the administration of a dual beta-lactam combination with two bacteriophages. The description is clear and detailed. The value of this work lies in the need of knowledge about clinical cases of patients infected with sequenced *M. abscessus* and treated with well-studied phages and an exhaustive in vitro analysis. However, there are some aspects that need to be improved or clarified:

1. In the Introduction section, it should be included the three subsp. of *M. abscessus* explaining that taxonomy is not in consensus right now (with references).

We have revised the Introduction (lines 60-67) addressing the three subspecies of *M. abscessus* and discussion on the classification.

2. It would be necessary to study the in vitro interaction between the antibiotics and the phages to know if a previous in vitro study could have been beneficial for the patient as a guide. It means to perform checkerboard assays or growth OD curves, for instance.

We have included additional data on synergy in Figure 6; a detailed response is provided below to a similar question from Reviewer #2.

3. It is said in the manuscript that there is no correlation between the effect of phages in vitro and in the patient, but I suggest to include in the Discussion section the following idea: in vitro studies usually are performed to kill bacteria in a planktonic state and no associated to a biofilm. Biofilm assays would be desirable to mimic better the real ambient of the bacteria and phage in the patient.

We thank the reviewer for this helpful comment. We have revised the Discussion (lines 332pp) commenting on the limitations of phage biofilm studies to date.

4. In the Discussion section it would be desirable to add some information about in vivo studies of phage therapy against *M. abscessus* and to comment if they are worthy or not.

We have included a comment in the Discussion that investigation of phage activity in animal model systems would be informative.

As a minor comment, I suggest to check the use of the past and present tense through the manuscript using the past for the results of the authors and the present for known information.

We have reviewed and revised where appropriate.

Reviewer #2 (Remarks to the Author):

This is a report of successful treatment of an extrapulmonary NTM infection of a sternal wound using a combined phage therapy/antibiotic approach. While the patient had both rough and

smooth *M. abscessus* infections and phages worked only against the former, the addition of dual beta-lactam therapy with meropenem and ceftazidime-avibactam resolved infection. The authors conclude that this therapeutic approach can be successful in complicated NTM skin and soft tissue infections.

This is a well-written and clearly presented study. The conclusions are generally well-supported by the data. While the targeting of NTMs with phages is not especially novel (this group has published much in this space) the context (a sternal wound) and the concomitant use of dual beta-lactam therapy are important and expand the scope of this therapeutic approach.

We thank the reviewer for the generally positive comments. We certainly recognize that this is not the first report of the use of bacteriophages for treatment of an NTM infection. The context has novelties as the reviewer notes, but one particular novelty not mentioned is the use of epigenetically modified phages to circumvent a host restriction. A major challenge in all of phage therapy but especially for NTM treatment is the relatively narrow host ranges, and there is much interest in the general aspect of how to address this. The strategy described here is the first report of one key solution which is to use epigenetically modified phages.

However, these data raise some questions:

1. Are there any evidence of synergy between the dual beta-lactam therapy and the phages used here against smooth NTM? The authors do not claim that there is but sensitizing effects of phages on otherwise antibiotic resistant bacterial isolates have been noted before and would be important to interrogate here.

This is a similar point raised by Reviewer #1. We agree that it would be helpful to know if there are synergistic interactions between the phages and the beta lactam antibiotics. But we note that this can only be readily tested using the smooth strain, where the phages do not kill very well. With the rough strain, the phages kill the bacteria efficiently over a wide range of phage/cell concentrations and MOIs as shown in the paper (if no synergism is noted with the smooth strain, then we agree that it would be worthwhile checking to see if the antibiotic antagonizes phage action in the rough strain). We agree that the possibility of synergistic interactions between the phages and the smooth strain is a question of interest, because in this case both rough and smooth strains were present clinically, and the resolution of the smooth strain was observed only after the beta lactams were administered. Although phage administration was stopped when the dual beta lactams were added, the phages may have persisted around the sites of infection and thus could have played a role clinically, if synergist is occurring.

We have now tested the synergy between the phages and the beta lactam antibiotics for the GD276B smooth isolate. The bottom line is that we do see synergy, with an increase in sensitivity to meropenem in the presence of either phage. The MIC is lowered by ~4-fold, with the effect diminishing as the phage concentration is reduced. Although we do not know if this behavior influenced treatment in this patient, it is certainly plausible, and the observation is of general importance for considering NTM therapies. We have added these observations to Fig. 6 and in a new supplemental figure and discussed them in the manuscript, including a new section in the Results (note that the legend for Fig. 5C was inadvertently included in Fig. 6, and we have corrected this).

2. Is there any evidence that elimination of one of the NTM isolates (the rough strain) made it easier to eliminate the other (the smooth strain) with conventional antibiotics? Polyclonal or polymicrobial infections can be more challenging to clear. It would be interesting to co-culture

the two NTM strains to see if this increases the concentration of antibiotic needed to eliminate these relative to treatment of each individually.

We have tested the MIC for Meropenem and for the MER/AVI/CAZ combination for the rough and smooth isolates separately as well as with the two strains combined. No differences were observed. This is an expected.

3. It would be important to discuss the necessity of the deep susceptibility testing that was done here. Given that the same phages (Muddy, Rosebud) have been used in several of the NTM treatment cases reported by this group are these studies in fact necessary once the presence of a rough strain is established? This is worth asking since the intensive testing regimens probably limit the more widespread and generic adaptation of this approach.

We have revised the Introduction to emphasize the point that currently phage therapies must be personalized.

REVIEWERS' COMMENTS

Reviewer #1 (Remarks to the Author):

The authors have answered all the required questions for reviewer #1 and no additional comments are required.

Reviewer #2 (Remarks to the Author):

The authors have addressed this reviewer's concerns.